# Changing the Contrast of Magnetic Resonance Imaging Signals using Deep Learning

**Attila Simkó**[1]                                                    ATTILA.SIMKO@UMU.SE
**Tommy Löfstedt**[2]
**Anders Garpebring**[1]
**Mikael Bylund**[1]
**Tufve Nyholm**[1]
**Joakim Jonsson**[1]

[1] *Department of Radiation Sciences, Radiation Physics, Umeå University, Sweden*

[2] *Department of Computing Science, Umeå University, Sweden*

## Abstract

The contrast settings to select before acquiring magnetic resonance imaging (MRI) signal depend heavily on the subsequent tasks. As each contrast highlights different tissues, automated segmentation tools for example might be optimized for a certain contrast. Unfortunately, the optimal contrast for the subsequent automated methods might not be known during the time of signal acquisition, and performing multiple scans with different contrasts increases the total examination time and registering the sequences introduces extra work and potential errors. Building on the recent achievements of deep learning in medical applications, the presented work describes a novel approach for transferring any contrast to any other.

The novel model architecture incorporates the signal equation for spin echo sequences, and hence the model inherently learns the unknown quantitative maps for proton density, $T1$ and $T2$ relaxation times. This grants the model the ability to retrospectively reconstruct spin echo sequences by changing the contrast settings Echo and Repetition Times. The model learns to identify the contrast of pelvic MR images, therefore no paired data of the same anatomy from different contrasts is required for training. This means that the experiments are easily reproducible with other contrasts or other patient anatomies.

Despite the contrast of the input image, the model achieves accurate results for reconstructing signal with contrasts available for evaluation. For the same anatomy, the quantitative maps are consistent for a range of contrasts of input images. Realized in practice, the proposed method would greatly simplify the modern radiotherapy pipeline. The trained model is made public together with a tool for testing the model on example images.

**Keywords:** Image reconstruction, Magnetic Resonance Imaging, MRI Contrast, Deep Learning, Unsupervised Learning, GAN

## 1. Introduction

Magnetic resonance imaging (MRI) is an essential step in the cancer treatment process for guidance in radiotherapy. The decision support solutions that are currently being developed and implemented often require certain contrasts. To expand the range of underlying data, tools already exist that change the domain of the solution (Zhu et al., 2018) or transferring the image to the desired contrasts (Dar et al., 2013) however the method usually transfers to only a single contrast which limits the usefulness of the method. We present a machine

learning method for transferring MRI data to custom selected contrasts in a manner that also shows quantitative information about the scanned anatomy.

The signal from a traditional spin echo sequence in MRI contains information about three quantitative properties of the scanned tissue: Proton density ($PD$), $T1$- ($T1$) and $T2$-relaxation times ($T2$). Their quantitativeness means that their values are physically meaningful and can be measured in physical units and compared between tissue regions and among anatomies. Due to the signal acquisition process, a single scan gives an image that is only a composition of these quantitative maps, as per the signal equation,

$$s = PD \cdot \left(1 - e^{-\frac{TR}{T1}}\right) \cdot e^{-\frac{TE}{T2}}, \tag{1}$$

where $TE$ and $TR$ stand for echo and repetition time, respectively, which are two of the settings for the given sequence, while all other settings are identical for the different contrasts. The two settings define the significance of the underlying maps in the signal, which is commonly used to categorize the contrasts into: $T2$-weighted ($T2w$), $T1$-weighted ($T1w$), and $PD$-weighted ($PDw$).

This work builds on the topicality of Deep Learning within medical applications. In particular, convolutional neural networks (CNNs) have achieved recent successes in tasks such as segmentation (Heller et al., 2019), super-resolution (Chen et al., 2018), and accelerated signal reconstruction (Zbontar et al., 2018), with solutions that are well-founded and groundbreaking. A Generative Adversarial Network (GANs, Goodfellow et al., 2014) is a non-cooperative game with two models that are trained simultaneously. Based on CNNs, the discriminator classifies an image as real or generated, and a generator is trained to produce images that will be classified as real by the discriminator. As the discriminator learns the distinctive characteristics of the different contrasts, the generator will also learn to simulate these differences. The strength of GANs have been showcased in recent research with Cycle-GANs where two models are trained to transfer from one domain of images to another and vice versa, examples include CT and MRI images, with state-of-the-art accuracy (Wolterink et al., 2017). They have been used to transfer between specific contrasts (Welander et al., 2018) however increasing the number of supported contrasts to transfer to also increases the complexity of the problem by the number of models to train. For easy reproducibility with any number of contrasts, we exclude CycleGANs as a candidate, but the unsupervised nature make GANs an ideal choice for the proposed task.

We introduce a novel conditional GAN architecture for contrast transfer of MRI data. To the best of our knowledge, this is the first work employing physical properties of the signal acquisition process to transfer MRI data between contrasts, achieving this novelty in an unsupervised fashion.

## 2. Materials and Methods

Pelvic MRI scans from 100 patients were captured with a 3T Signa PET/MR scanner (GE Healthcare, Chicago, Illinois, United States) at the University Hospital of Umeå, Sweden (ethical approval nr. 2019-02666). To explore the characteristics of the different contrasts, the sequences used five different $TE$ and $TR$ combinations, covering a large span of values, collected in Table 1.

Table 1: The five combinations of $TE$ and $TR$ that were used to acquire the dataset. The contrasts from left to right constructed: two $T2w$, two $T1w$, and one $PDw$ signals.

|          | $T2w$ |      | $T1w$ |     | $PDw$ |
|----------|-------|------|-------|-----|-------|
| $TE$ [ms] | 75    | 120  | 8     | 8   | 8     |
| $TR$ [ms] | 4500  | 4500 | 400   | 750 | 4500  |

Given the self-supervising aspect of the devised method, no multi-contrast scans were required for training, therefore each data sample in the *training dataset* contains a scan from only a single contrast and its corresponding $TE$ and $TR$ values. The dataset contains scans from 90 patients. To speed up the data acquisition process, every patient was scanned for all 5 contrasts independently, however to ensure there are no registered slices in the datasets, all slices were shifted randomly in both directions by up to 10 pixels.

The *validation dataset* contained multi-contrast sequences from five patients, where all slices were available in all five contrasts, corrected for possible patient movement by non-rigid registration. Each data sample contained an input scan, paired with a target scan of the same anatomy but from all five available contrasts. This resulted in 1,875 slice pairs with each sample also containing the $TE$ and $TR$ values of the target contrast.

A *test dataset* was created identically to the validation dataset, using five different patients.

*Generator:* The network architecture was based on the U-Net (Ronneberger et al., 2015), where instead of a convolutional layer returning the map that should correspond to the target signal, the final convolutional layer returned three $256 \times 256$ maps. These were used to construct the output layer as in Equation 1 using the $TE$ and $TR$ values of the target contrast, therefore each map uniquely defined $PD$, $T1$, and $T2$. By mirroring the signal equation, the model reflects the underlying physics of the task. The maps for $T1$ and $T2$ were clipped above the value 10 and 5, respectively (values in seconds), to exclude values that never occur in patients (Bojorquez et al., 2017; Stanisz et al., 2005), and as a form of regularization to help the training. The quantitative maps were neither known nor needed for the training process, since the output of the model is the reconstructed signal $s$, however they should be consistent despite the contrasts of the input, and they should agree with values from literature. The input for the generator was the input image and the $TE$ and $TR$ values of the target contrast.

*Discriminator:* The task of the discriminator was to classify images as fake or real, and their specific contrast. The network's architecture was based on that of Salimans et al. (2016). The discriminator for our task had $5 + 1$ output classes: 5 to classify the image contrast and 1 to classify the image as fake. The architecture was a PatchGAN (Isola et al., 2017), where instead of obtaining a class for an input image, the classifier returns a map of classes from different patches of the input image. This allowed the discriminator to detect smaller differences, while simultaneously stabilizing the training. The discriminator settings looked at patches of $190 \times 190$ (which meant an output map of $8 \times 8$), avoiding patches containing only the background. A final softmax layer ensured that in each element of the map, only one output class was selected by the discriminator.

*Training process:* The mean absolute error loss was used for training with the Nadam optimizer (Dozat, 2016), with a learning rate of 0.0005. Both networks were updated in every training iteration, and the performance of the generator was evaluated for contrast transfer at the end of each epoch using the mean squared error (MSE) metric.

Further details about the models, and the training process can be found in the appendix.

## 3. Experiments

We evaluate the model for contrast transfer followed by further experiments to show the functionality of the proposed approach. This includes evaluating the quantitative maps that are generated by the model, and investigating how changing the contrast settings affects the discriminator's performance.

### 3.1. Contrast transfer

For an overall evaluation, the model was used on each contrast of every slice in the testing dataset, to predict the same slice from every other contrast. Together with visual assessment, the evaluations used MSE, normalized root-mean-squared-error (NRMSE), peak signal-to-noise ratio (PSNR), and the structural similarity index (SSIM). The overall prediction error was further investigated, splitting by input and target contrasts.

The difference of the original images from all the combination of contrasts was computed and reported below as the baseline error (if the generator would output the input images without modifying them, the model would achieve the baseline error). The error was only calculated for the anatomy, excluding the background noise using Otsu thresholding (Gonzalez and Woods, 2006).

### 3.2. Quantitative maps

To obtain ground truth for the underlying quantitative maps of the signal, we used all five contrasts and their contrast settings to approximate the maps using the least-squares method. We used the Levenberg-Marquardt algorithm to minimize the least squares error. Likely due to the amount of outliers in the quantitative data, and possibly due to noise and registration errors, for reconstructing the overall signal from the recovered maps, the errors were large. Instead the method was used only to approximate the quantitative maps for four manually segmented tissues, namely: fat, muscle, bladder, and the prostate. Each tissue class in the test dataset contained 109,308, 154,588, 9,639, and 3,129 voxels, respectively. The segmentations were performed using MICE Toolkit[1] (NONPI Medical AB, Umeå, Sweden, Nyholm and Jonsson, 2014).

The maps from the least-squares approximation for the segmentations were used as ground truths when evaluating the quantitative maps obtained from the trained model. The mean intensities of the quantitative maps from the least-squares method were compared to the results from the model, and to values found in literature (Bojorquez et al., 2017; Stanisz et al., 2005).

---

1. https://www.micetoolkit.com/

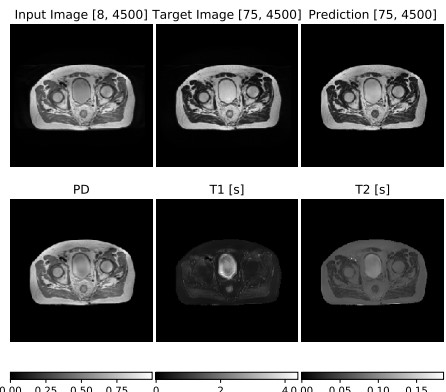

Figure 1: A randomly selected example showing the transfer from $PDw$ contrast ('Input Image') to $T2w$ ('Target Image') by plotting the corresponding prediction of the model ('Prediction'). The bottom row shows the quantitative maps acquired from the input image.

### 3.3. Decision boundaries

The performance of the generator relies heavily on the discriminator. Although the discriminator was only trained on five different contrasts, it is expected to work for other $TE$ and $TR$ combinations as well, classifying the contrast as the class it is most similar to. The discriminator is expected to be most accurate for classifying images that were included in the training dataset, and interpolating between these values should show a smooth transition in prediction accuracy.

Two cases were investigated, one for the $T1w$ and $PDw$ contrasts, which have the same $TE = 8ms$, the $TR$ values were interpolated to cover all three available values. The other case is for the $T2w$ and $PDw$ contrasts, that share $TR = 4500ms$, and therefore the $TE$ values were interpolated. For each interpolated contrast we generated synthetic signal using the entire test dataset. Then the most significant class from the $8 \times 8$ maps predicted by the discriminator was marked by the corresponding color on the map. The confidence in the class predictions are illustrated by the darkness of the color. The predictions should correlate with both $TE$ and $TR$.

## 4. Results

The lowest validation error was achieved in epoch 390 of the training process, and therefore the model from this epoch was selected for further evaluation.

### 4.1. Evaluating contrast transfer

Visualization of the performance of the model is presented on Figure 1. It shows the results for transferring a randomly selected slice from a single contrast to another while also showing the quantitative maps acquired before constructing the output signal. Four other examples are collected in the appendix on Figure 7.

For a quantitative evaluation, the experiment investigated how well the model transferred from each contrast to all other contrasts and the results are collected in Table 2.

Table 2: The results for contrast transfer. The 'Baseline' error is calculated from the original images, and 'Model' shows the errors for the predictions of the trained generator.

|  | Baseline | Model |
|---|---|---|
| MSE | $0.010 \pm 0.007$ | $0.004 \pm 0.002$ |
| NRMSE | $0.215 \pm 0.080$ | $0.144 \pm 0.041$ |
| PSNR | $21.53 \pm 3.93$ | $25.05 \pm 2.51$ |
| SSIM | $0.961 \pm 0.019$ | $0.974 \pm 0.011$ |

All metrics reached the same conclusions when plotted against input and target contrasts, therefore only the NRMSE results are presented. In Figure 2 each row and column shows the baseline and prediction errors for the corresponding input and target contrast. The diagonal shows the error of generating the contrast of the input image.

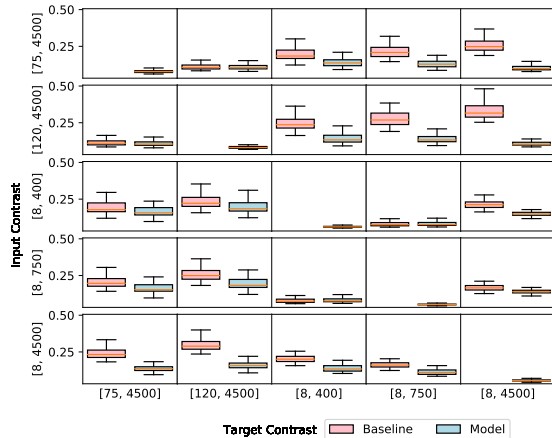

Figure 2: The distribution of baseline and prediction NRMSE between input and target contrasts. The labels for the contrasts mean their $TE$ and $TR$ values in $ms$. The sum of all values is presented in Table 2 as the 'Baseline' and 'Model' errors.

### 4.2. Evaluating quantitative maps

The mean values of the quantitative maps from the two methods are collected in Table 3.

### 4.3. Evaluating decision boundaries

The top plots in Figure 3 illustrates where the original training data images fall on the two maps. The left figures show the results of interpolating $TR$ with $TE = 8ms$, while the right plots show the results of interpolating $TE$ with $TR = 4500ms$. Only five combinations are shown, following the contrasts from left to right in Table 1, they are: blue, green, orange, purple, and red. The bottom plots show the results after generating contrasts for extended $TE$ (left) and $TR$ combinations (right).

Table 3: The predicted quantitative maps using a least-squares approximation (LSQ) and using the trained generator (Model) on the testing dataset.

| | $PD$ | | $T1$ [s] | | $T2$ [s] | |
|---|---|---|---|---|---|---|
| | LSQ | Model | LSQ | Model | LSQ | Model |
| Fat | $0.60 \pm 0.09$ | $0.58 \pm 0.09$ | $0.38 \pm 0.07$ | $0.43 \pm 0.07$ | $0.21 \pm 0.05$ | $0.06 \pm 0.01$ |
| Muscle | $0.31 \pm 0.08$ | $0.30 \pm 0.06$ | $0.76 \pm 0.19$ | $0.72 \pm 0.17$ | $0.07 \pm 0.03$ | $0.04 \pm 0.01$ |
| Bladder | $0.54 \pm 0.16$ | $0.38 \pm 0.13$ | $3.07 \pm 0.90$ | $1.97 \pm 0.78$ | $2.71 \pm 1.89$ | $0.12 \pm 0.04$ |
| Prostate | $0.43 \pm 0.04$ | $0.37 \pm 0.08$ | $1.20 \pm 0.16$ | $1.34 \pm 0.52$ | $0.11 \pm 0.02$ | $0.06 \pm 0.03$ |

## 5. Discussion

The evaluation of the contrast transfer (Table 1) shows an improvement over the baseline error, with the specific improvements further examined in Figure 2. They both indicate that the discriminator learns about the contrast and helps the generator to reconstruct these differences by mapping the features of the input images into $PD$, $T1$, and $T2$ maps. Looking at how the presented errors distribute across input and target contrasts, we first note that the prediction errors for reconstructing the input contrast are expectantly worse than the baseline error, which is zero (on the diagonal). Except for the diagonal, all errors from the proposed model are below their corresponding baseline error. The smallest improvements are for transferring between similar contrasts ($T1w$ to $T1w$ or $T2w$ to $T2w$) where the baseline error is also small. For any input contrast, the baseline error changes substantially based on the target contrast, however this change is decreased for the prediction errors of the model, showing that they are less dependent on the target contrast.

Evaluating the quantitative maps in Table 3 shows that the $T1$ and $T2$ values from LSQ agree with common values from literature without substantial differences. However, the standard deviations for the bladder are large for all three maps showing inhomogeneity of the tissue and possible registration errors. The $PD$ maps can not be compared to values from literature, as they have an arbitrary scale, but the values are still comparable between the two methods for the same dataset. The results illustrate that the proposed model predicts similar maps as the LSQ method for $PD$ and $T1$, while the $T2$ values are generally substantially smaller in the maps from the model, especially for the bladder. The reconstructed $T2$ maps suggest that the accuracy of the model will be limited for contrasts that rely heavily on accurate $T2$ maps, however for such a case, transferring from a short

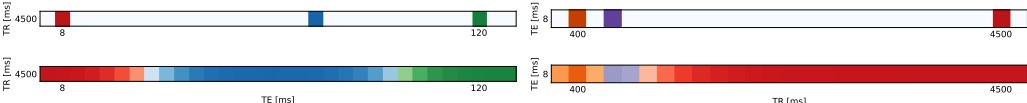

Figure 3: *Top:* The map shows the predictions of the discriminator on the test dataset that only contains five different contrasts. The five classes are distinct and the dark color shows the certainty of the predictions. *Bottom:* The generator was used to extend the possible $TE$ and $TR$ combinations and the map shows the predictions of the discriminator.

$TE\ PDw$ contrast to $T2w$, Figure 1 shows good similarities for the reconstructed signal most noticeably from the bladder and muscle, and the reconstruction errors for such cases (Figure 2, bottom row, first two columns) show improvement over the baseline error. This result, together with the low standard deviations for the $T2$ maps suggest that the model can still accurately reconstruct signal from the selected contrasts in the dataset without improving the results for the $T2$ maps. This implies that training the model on more data (and not necessarily a wider range of contrasts), the overall prediction error would decrease, focusing on smaller changes between the contrasts, making the $T2$ maps more accurate.

The maps created by the extended $TE$ and $TR$ combinations in Figure 3 illustrates that the decision boundaries strongly correlate with both $TE$ and $TR$ between the values that were included in the training dataset, showing a smooth transition. This further supports that the discriminator is working as intended.

## 6. Conclusions

The proposed architecture changes the essence of the problem by instead of transferring to another contrast, the signal is decomposed into the three quantitative maps, and then reconstructing a signal of the desired contrast. The results corroborate the effectiveness and the vast possibilities of machine learning in medical imaging as the model is able to perform a consistent decomposition of the signal without knowing *anything* about the quantitative maps during training. The contrasts included in the training need to be selected in a systematic way to cover a large span of $TE$ and $TR$ and a wide range of their combination. If this is not done, the results are only expected to work well in the parameter space spanned by the contrasts included in the training process. Since no multi-contrast scans are required, the training dataset is easily reproducible and can be expanded for other number of contrasts and anatomies as well.

The results of the presented study show that retrospective reconstruction of MRI signal using custom contrast settings is possible. A venue for future research would include collecting data with a larger range of contrasts and expanding the model with other sequences and anatomies as well. With continuous evaluation and careful supervision, an effective application example of the method is in radiotherapy, when performing another scan with different settings might not be possible.

The model evaluated in this paper is made publicly available[2] (in a .h5 format) for further use, together with a tool for testing the network. To help with testing, three example images were also published, taken with the same protocol as the images in the training dataset.

## Acknowledgments

We are grateful for the financial support obtained from the Cancer Research Foundation in Northern Sweden (LP 18-2182, AMP 18-912, AMP 20-1014), the Västerbotten regional county, and from Karin and Krister Olsson. The computations were performed on resources provided by the Swedish National Infrastructure for Computing (SNIC) at the High Performance Computing Center North (HPC2N) in Umeå, Sweden, partially funded by the Swedish Research Council through grant agreement no. 2018-05973.

---

2. http://doi.org/10.5281/zenodo.4530894

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

## Appendix A. Network Architectures

The generator was built on the U-Net (Ronneberger et al., 2015) architecture. The architecture outputs three features maps, that are passed through a custom layer implementing Equation **??**. *I.e.*, the feature maps must implicitly learn the $PD$, $T1$, and $T2$ maps. See Figure 4 for an illustration of the generator architecture.

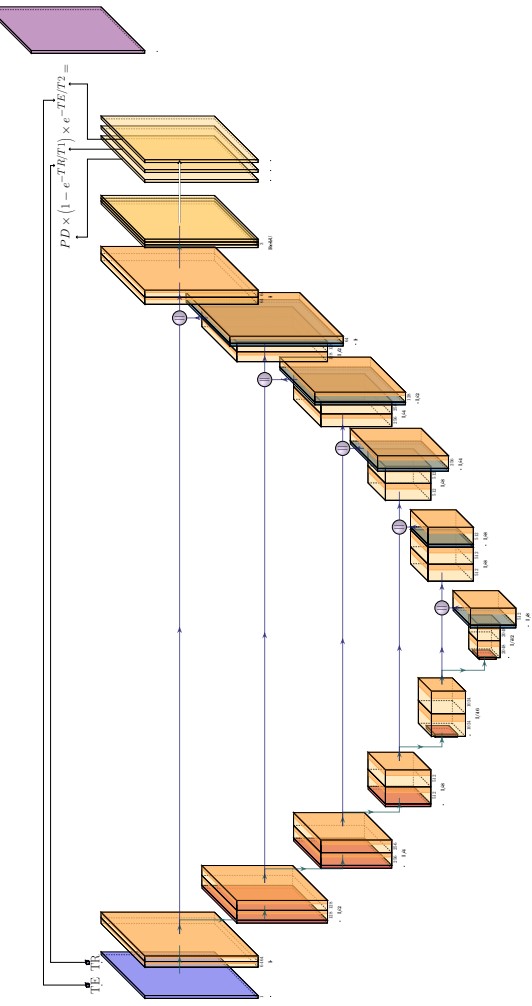

Figure 4: The U-Net-based architecture for the generator. The inputs (all blue) are the slice on the bottom and the $TE$ and $TR$ values of the target slice (all blue). The outputs of the U-Net architecture are three feature maps that are split and used in the signal equation as $PD$, $T1$, and $T2$. As regularization, the maps for $T1$ and $T2$ were constrained to a maximum value of 10 and 5, respectively, corresponding to box constraints on $[0, 10]$ and $[0, 5]$, respectively corresponding to seconds. Using the input $TE$ and $TR$, the output of the generator is the reconstructed signal (violet).

The discriminator was a $190 \times 190$ PatchGAN. The output was an $8 \times 8$ map classifier of six output classes: the image being from one of the five contrasts, or fake. See Figure 5 for an illustration of the discriminator architecture.

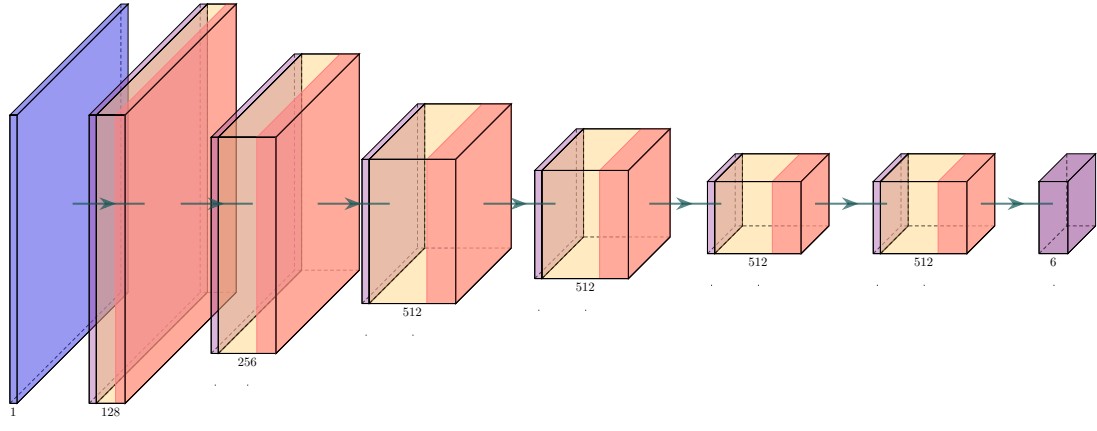

Figure 5: The classifier used as the discriminator was a PatchGAN with patch size $190 \times 190$. The input on the left (blue) is the image slice to be classified. The discriminator contained six convolutional blocks, each starting with adding Gaussian noise with a standard deviation of 0.0005, then followed by a convolutional block, with kernel size 3 and LeakyReLU activations with $\alpha = 0.2$. The output layer (violet) used a softmax activation, where each element is a contrast classifier on a $190 \times 190$ patch of the input image.

## Appendix B. Training process

In the first stage, a simple pre-training was employed on both the generator and the discriminator, separately. For the generator such that it returned the input image despite the contrast settings, and for the discriminator to achieve an 80 % accuracy when classifying the real images by contrast, *i.e.*, without yet introducing generated images. In the second stage, the generated images were introduced for training the discriminator, and the generator was then only updated through the discriminator using a mean absolute error loss.

During the training, the MSE was monitored on the validation dataset for insight. Training for 500 epochs took approximately 4 days using an Nvidia GTX 2080 Ti card. The network reached the lowest validation scores in epoch 390 which was therefore selected for evaluation. Figure ?? illustrates how the accuracy for labeling the fake images progressed during training. During pre-training the generator learned to output the input image by placing all information in the proton density maps while making $T1$ and $T2$ constants, which can be seen at the first mark of training (epoch 10) in Figure ??. If the training parameters of a GAN are selected carefully, the performance of the discriminator and the adversarial network stay even while the generator improves. Although the accuracy of the adversarial network decreases in time, it never becomes and stays zero, which would mean that it was dominated by the discriminator. Two other marks were added to visualize how the network improved during training: At epoch 100 and at epoch 390. The final mark shows results from the model selected for further evaluation.

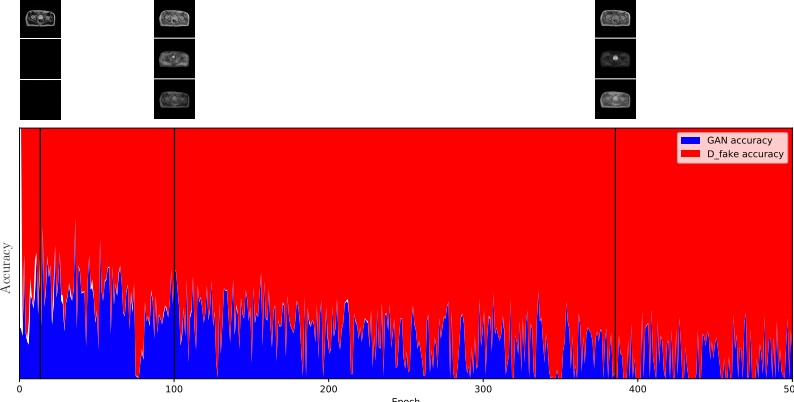

Figure 6: The accuracy of classifying the generated images during training shows the non-cooperative game of GANs. After each epoch the performance of the discriminator network is evaluated through the accuracy of the discriminator for classifying generated images as fake (red line, `D_fake_accuracy`), while the performance of the adversarial network was evaluated by the accuracy of the discriminator for classifying generated images as the contrast they belong to (blue line, `GAN_accuracy`). The white lines show where images were classified as real but from the incorrect contrast. For each epoch the length of these three lines add up to 100%. The figure shows example results from three epochs (from left to right: 10, 100, 390) and from each epoch the quantitative maps from a sample image (from top to bottom: $PD$, $T1$, $T2$).

## Appendix C. Additional Images

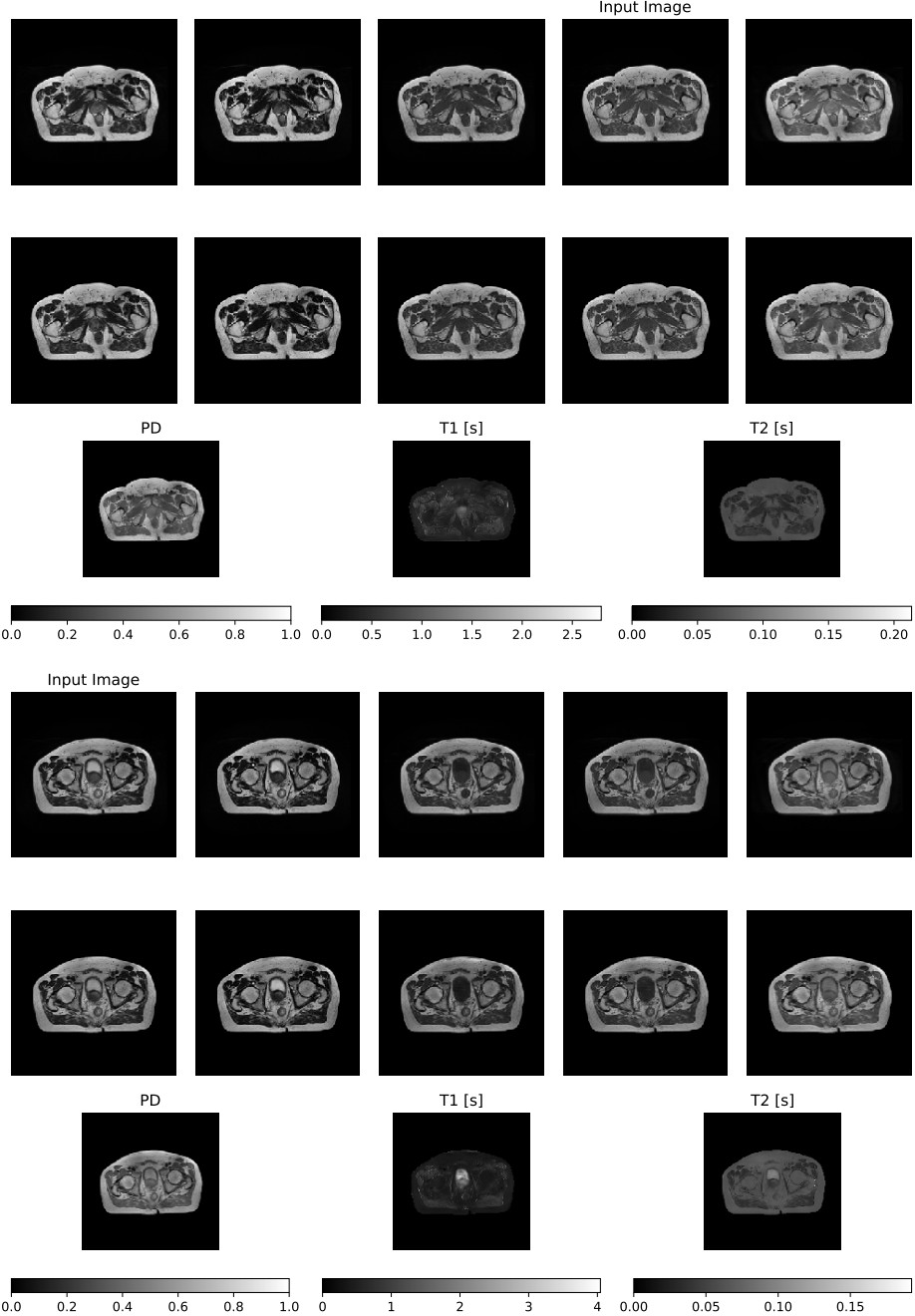

Figure 7: For example cases of the original contrasts (top), and the corresponding output contrasts transferred from the 'Input Image'. The bottom row shows the predicted quantitative maps: $PD$, $T1$, and $T2$.

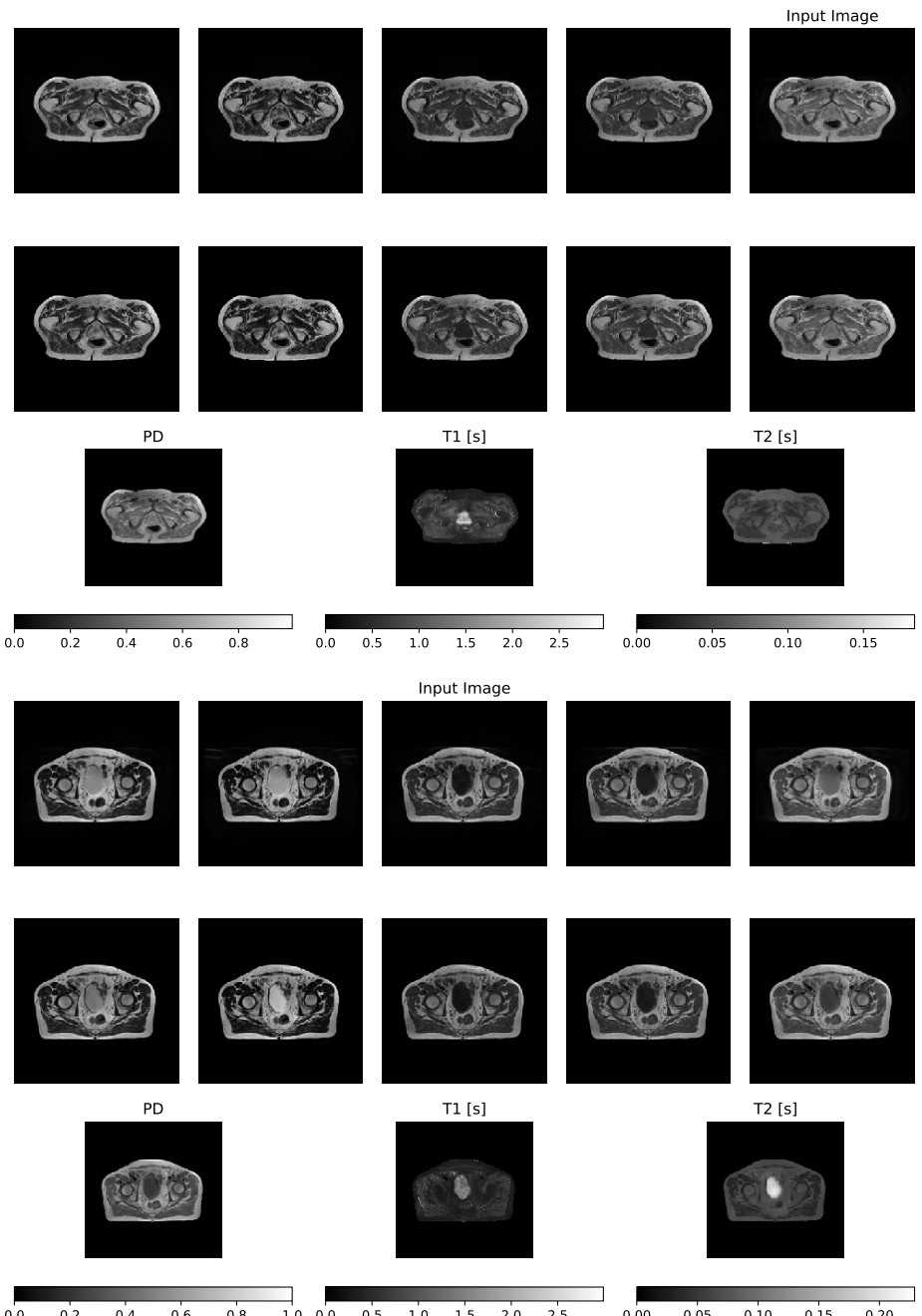

Figure 7: For example cases of the original contrasts (top), and the corresponding output contrasts transferred from the 'Input Image'. The bottom row shows the predicted quantitative maps: $PD$, $T1$, and $T2$. (cont.)

