# OpenReview forum: "Changing the Contrast of Magnetic Resonance Imaging Signals using Deep Learning"
_MIDL.io/2021/Conference — MIDL 2021_

### Official Review · AnonReviewer2 · 2021-03-08

**Confidence:** 4
**Preliminary Rating:** 2
**Final Rating:** 2

**Summary:**

The authors proposed a method to transfer MRI images to images of different contrasts. The method aims to learn the quantitative information (PD, T1, T2) from the input MR image in an unsupervised manner using a GAN-based approach. The generator of GAN is based on U-net, mapping the input image to three quantitative maps that are used for reconstructing the target image using the spin-echo signal equation given known TE and TR values.  The discriminator classifies the target whether it is fake or it belongs to one of the five contrast categories with different TE-TR combinations. The learned quantitative maps enable reconstructing MR images of various contrasts from an input MR using different TE and TR values. In the experiments, the proposed method was shown to be superior to a baseline and provides similar quantitative information to that obtained from least-square. The authors also showed that the trained discriminator has a smooth decision boundary in the TE-TR parametric space.

**Strengths:**

- Instead of directly learning the images of target contrast, the proposed method learns the quantitative information from the input MR image, which theoretically enables reconstructing MR images of any contrasts given different set of TE and TR values. This is particularly useful for scenarios that require multiple scans of different contrasts.
- Leaning the quantitative maps does not require the ground truth of PD, T1 and T2, which was realized by adversarial learning.

**Weaknesses:**

Overall, the evaluation of the method is not convincing enough to me.

- Evaluation on contrast transfer
	- In Table 2, the metrics of Model indeed looks better than Baseline, but what does those numbers mean in actual practice? Do they mean the generated images look almost the same as the real images? Do they mean that the generated images are good enough for transferring to clinical use?
	- In Table 2, is the proposed method always better than the baseline for different contrasts (with different TE/TR values)? It would be nice to show the metrics for different contrasts.
	- The baseline seems not very strong (given that if the generator does not change the input images), the advantage of the proposed method over the baseline should be obvious. Therefore, it's hard to draw a conclusion if the proposed method is truly good or not.
	- In Figure 1, is there a particular reason why only the distribution of NRMSE is shown but not the other metrics? It would also be good to show the metrics of the baseline in the boxplot for comparison.
- Evaluation on quantitative maps
	- The ground truth were obtained using a least-square approximation. Is this a good ground truth to be compared with?
	- In Table 3, for T2, the Model is not so similar to LSQ, does it mean if the model is not good enough for predicting T2 or if the ground truth is not good enough? Is there any reason why the model and LSQ are not so similar on T2?
- Evaluation on decision boundaries
	- Although it's generally nice to show such a decision boundary to provide some insights into the discriminator, the validity of this map is questionable. In Figure 2 (left), the sampled contrasts are generally on the upper or right side of the parametric space, are they representative enough for any other points in this map? For example, for a contrast on the lower left corner of the map (with high TE and low TR), does the image look similar to any of the five contrast images? Should this point belong to one of the five contrast classes that the discriminator can predict?

**Deanonymize Review:**

no

**Detailed Comments:**

- Please explain more clearly how to obtain the final class of the input image from the 8x8 output map of the discriminator?
- In Figure 2, there is no black point or area as described by section 4.3.
- In the last paragraph of page 7, the author wrote "PD maps can not be compared to ground truth, ... But the values are still comparable between the two methods for the same dataset ... " Are they comparable or not?
- In the second paragraph of page 8, " ... the decision boundaries strongly correlated with TE and TR ..." The boundaries seems not strongly correlated with TE or TR in Figure 2 (right), for example, with large TE values, TR almost has no influence on the decision boundaries.

**Final Rating Justification:**

I would like to thank the authors for answering my questions in the rebuttal and updating the paper accordingly. Although most of my questions have been addressed, there are two main concerns remaining regarding to the choice of a weak baseline and whether LSQ serves as a good ground truth (please see my comments below for details), which weaken my confidence on the acceptance of the paper.

**Justification Of The Preliminary Rating:**

The idea of learning the quantitative information from MR images in an unsupervised manner, which potentially enables generating MR images of any contrasts, is interesting. However, the validation of the method is below sufficient, which undermines the paper.

**Paper Type:**

methodological development

**Questions To Address In The Rebuttal:**

- Please address the questions and comments in the weaknesses.
- In the conclusion, the authors claim that the training requires a large span of TE and TR and a wide range of their combination. In this paper, does the selection TE and TR values cover enough in the parametric space? If a large span of TE and TR is not available, can the trained model still generate MR images of any contrasts? These questions are a bit related to my comments in the weaknesses about the evaluation on decision boundaries.

**Special Issue:**

no

---

> ### Author Response · Authors · 2021-03-16
> **Response to review #3 1/2**
>
> "In Table 2, the metrics of Model indeed looks better than Baseline, but what does those numbers mean in actual practice? [...] Do they mean that the generated images are good enough for transferring to clinical use?"
>
> A visual assessment was added to the main text to get a grasp of how the model performs.
> The expanded boxplot shows a better distribution of the errors between input and target contrasts. The mean predicted error is always smaller than the baseline error except for the diagonal, where the baseline error is 0 and decomposing the signal into the quantitative maps and then using the original TE and TR values does not reconstruct the original image "perfectly''. As the revised Discussion states ``For any input contrast, the baseline error changes significantly based on the target contrast, however this change is decreased for the prediction errors of the model, showing that they are less dependent on the target contrast.''
> The authors are convinced in the functionality of the presented method, but transferring to clinical use was outside the scope of the current project.
>
>
> "In Table 2, is the proposed method always better than the baseline for different contrasts (with different TE/TR values)? [...]"
>
> The authors agree and expanded the boxplot to include the baseline errors similarly. The mean predicted error is smaller than the baseline error for all cases except for the diagonal, where the baseline error is 0.
>
>
> "The baseline seems not very strong (given that if the generator does not change the input images), the advantage of the proposed method over the baseline should be obvious. Therefore, it's hard to draw a conclusion if the proposed method is truly good or not."
>
> To better show the performance of the model, a visual assessment of the method was added to the paper, while the boxplot evaluations were updated to contain the baseline errors as well. The prediction errors were more similar for the different target contrasts than the baseline errors, meaning that the prediction error is less dependent on the target contrast. We can also see that for the target contrast which is the least similar to the input, the predictions have the most significant improvement over the baseline error. For a more general evaluation of the method, we can turn to the evaluation of the quantitative maps.
>
>
> "In Figure 1, is there a particular reason why only the distribution of NRMSE is shown but not the other metrics? [...]"
>
> The authors updated the paper with the relevant missing information that ``The conclusions drawn from plotting the errors by input and target contrasts were identical for all metrics, therefore only the NRMSE results are presented''. The boxplot is updated with the baseline error for each input-target contrast combination.
>
>
> "The ground truth were obtained using a least-square approximation. Is this a good ground truth to be compared with?"
>
> Although the Gaussian assumption needed for least-squares approximations is somewhat violated, and the three variables are approximated using only five equations, and thus the approximations could be biased, however as stated in the updated Discussions, "the T1 and T2 values from LSQ agree with common values from literature without substantial differences'' which indicates that the results of the method are reliable.
>
>
> "In Table 3, for T2, the Model is not so similar to LSQ, does it mean if the model is not good enough for predicting T2 or if the ground truth is not good enough? Is there any reason why the model and LSQ are not so similar on T2?"
>
> The difference between the LSQ and the model T2 maps are now discussed in more detail in the revised Discussion the paper, ``Transferring from a short TE, PDw contrast to T2w images rely heavily on accurate T2 maps, however the reconstruction errors show improvement over the baseline error, which together with the low standard deviations for the T2 maps suggest that the model can still accurately reconstruct signal from the selected contrasts in the dataset without improving the results for T2. This implies that training the model on more data (and not necessarily a wider range of contrasts), the overall prediction error would decrease, focusing on smaller changes between the contrasts, making the T2 maps more accurate.''

---

> > ### Comment · AnonReviewer2 · 2021-03-20
> > **The choice of the baseline method and LSQ as the ground truth**
> >
> > Thank the authors for answering my questions and updating the manuscript. There are still two main concerns on the evaluation of the proposed method which I found not well addressed.
> > - The baseline method is very weak, which is merely the case as if the generator outputs the input image without making any change. It should not be difficult to beat such a weak baseline. Therefore, compared to this baseline method, it is hard to judge whether the proposed method is truly strong or not. A stronger baseline method for comparison would be appreciated, e.g. train Cycle-GAN to transfer each pair of the five contrasts as a reference method.
> > - I'm still not convinced enough that LSQ is a good "ground truth". The authors wrote "the T1 and T2 values from LSQ agree with common values from literature without substantial differences". What are the common values from the literature? What are "substantial differences"? Could the authors directly show the values from literature in the paper and let the readers judge the difference? Additionally, the authors wrote in the updated manuscript "The PD maps can not be compared to values from literature ...", then how should one believe if LSQ gives a good ground for PD?

---

> ### Author Response · Authors · 2021-03-16
> **Response to review #3 2/2**
>
>
> "Although it's generally nice to show such a decision boundary to provide some insights into the discriminator, the validity of this map is questionable. In Figure 2 (left), the sampled contrasts are generally on the upper or right side of the parametric space, are they representative enough for any other points in this map? For example, for a contrast on the lower left corner of the map (with high TE and low TR), does the image look similar to any of the five contrast images? Should this point belong to one of the five contrast classes that the discriminator can predict?"
>
> The authors agree completely with the reviewer that this experiment only shows that the transition is smooth between the explored contrasts included in the training dataset. This is what the discriminator was trained on, therefore although the generator is expected to work outside of this parameter space, the discriminator is not. The results are only meaningful from a subset of the maps presented originally, which was implied in the original paper as "the extended TE and TR combinations [shows] a smooth transition between the values included in the training dataset". The authors revised the experiment to exclude the part of the map that does not contain meaningful information. The authors believe that with more contrasts included in the training dataset, a larger map could be meaningfully explored, however for the current case the experiment can be simplified to still contain all relevant information as the original. The first is when TE is fixed for 8ms, and TR is interpolated between 400-4500ms, this explores the decision boundaries between the T1w and PDw contrasts. Then for the other case we fix TR for 4500ms and interpolate TE between 8-120ms. The two cases cover the correlation of the discriminator with both TE and TR. This separation hopefully excludes all insignificant parts of this experiment. The authors would like to thank the reviewer for the insight, and are grateful for the comment, improving the quality of the paper.
>
>
> "Please explain more clearly how to obtain the final class of the input image from the 8x8 output map of the discriminator?"
>
> The authors revise the description of the discriminator to clarify that "instead of obtaining a class for an input image, the classifier returns a map of classes from different patches of the input image.'' For the boundary decision experiment the displayed "class'' for a specific TE and TR combination is obtained as the most common value from the maps of all generated images of the testing dataset of all five contrasts. How common this value was is displayed by the darkness of the color.
>
>
> "In Figure 2, there is no black point or area as described by section 4.3."
>
> The main text was updated to refer to the correct color, "orange''
>
>
> "In the last paragraph of page 7, the author wrote "PD maps can not be compared to ground truth, [...]"
>
> The authors apologize for the error, the ground truth in the original paper referred to the ground truth from literature and not the LSQ approximations. The Discussion was updated to clarify that "PD maps can not be compared to values from literature, since they also contain the bias, and other scalar terms that can depend on the scanner type, and so the maps have an arbitrary scale''.
>
>
> "[...]The boundaries seems not strongly correlated with TE or TR in Figure 2 (right), for example, with large TE values, TR almost has no influence on the decision boundaries."
>
> The authors revised the experiment and its conclusions, and instead state that the ``the decision boundaries strongly correlate with both TE and TR on between the values that were included in the training dataset, showing a smooth transition''. The authors believe that by using more contrasts during the training phase, the correlation of the decision boundaries with TE and TR could be further explored. The authors believe this revision greatly improves the evaluation of the discriminator.
>
> "In the conclusion, the authors claim that the training requires a large span of TE and TR and a wide range of their combination. In this paper, does the selection TE and TR values cover enough in the parametric space? If a large span of TE and TR is not available, can the trained model still generate MR images of any contrasts? "
>
> The conclusions were updated regarding the need to cover a large span of TE and TR combinations stating that "If this is not done, the results are only expected to work well in the parameter space spanned by the contrasts included in the training process''. However as the updated Discussion now states, the authors believe that although the current project would benefit from training on more data "(and not necessarily a wider range of contrasts), the overall reconstruction error would decrease, focusing on smaller changes between the contrasts, making the T2 maps more accurate''.

---

### Official Review · ~Matthan_W._A._Caan1 · 2021-03-08

**Confidence:** 5
**Preliminary Rating:** 3
**Recommendation:** Poster

**Summary:**

The paper proposes a GAN-based model architecture, using a U-Net for the generator, to generate quantitative maps for PD, T1, and T2 relaxation times. The model can retrospectively reconstruct quantitative maps without ground truth data by incorporating the signal equation for spin-echo sequences and changing the TE and TR.

**Strengths:**

The paper shows the strengths of using DL methods for domain transfer learning, introducing a novelty of generating quantitative maps by incorporating the signal acquisition equation. The method is promising to extend on a broader range of TE and TR values and anatomies.

**Weaknesses:**

While the error in reconstructing T1-images is small, T2-errors are large, for all tissues studied. This is also e.g. visible in Fig 6, second subject, right image, where T2w-contrast is lacking in the center-top-part. This raises the question if T2w-contrast can be inferred from short TE T1w-images. Please discuss as a limitation of this work.

Minimal prior work on the field is mentioned, domain transfer learning approaches have been proposed before [1, 2]. The novelty of incorporating the signal acquisition equation and using unsupervised learning should be evaluated against relevant work.
1.     Dar, SUH, Özbey, M, Çatlı, AB, Çukur, T. A Transfer‐Learning Approach for Accelerated MRI Using Deep Neural Networks. Magn Reson Med. 2020; 84: 663– 685. https://doi.org/10.1002/mrm.28148
2.     Knoll, F, Hammernik, K, Kobler, E, Pock, T, Recht, MP, Sodickson, DK. Assessment of the generalization of learned image reconstruction and the potential for transfer learning. Magn. Reson. Med. 2018; 81: 116– 128. https://doi.org/10.1002/mrm.27355

The authors can be more precise in their terminology of multi-contrast data through the text. It is mentioned that the train and val sets were based on multi-contrasts scans (from all five contrasts), followed by the remark that given the self-supervising aspect of the method, no multi-contrast scans were required for training.


**Deanonymize Review:**

yes

**Detailed Comments:**

Fig. 2 represents a contrast in orange color, whereas in the text it is mentioned as black.

The publicly available materials provide a viewer for three given images instead of, e.g., gifs, with the option to change contrast settings. There is no real value of the publicly available materials since no code is available, neither can be extended on other data using the pretrained model.

Fig. 6 is hard to interpret, please reduce margins. The second row should be replaced by plotting the difference between the target and the predicted image. The bottom row should contain pairs of the predicted quantitative maps and the LSQ ground truth maps. Also adjust the range, removing peak intensities, and consider saturating the bladder (T1/T2 are difficult to estimate in free water)


**Justification Of The Preliminary Rating:**

This paper exploits knowledge of the MR physics model in transfer learning, for the purpose of radiotherapy. The method appears to perform only in part of parameter space, excluding the clinically relevant T2 weighting/mapping, which is a limitation of this work. For T1w-images, the results are convincing.

**Paper Type:**

methodological development

**Special Issue:**

no

---

> ### Author Response · Authors · 2021-03-16
> **Response to review #2**
>
> "While the error in reconstructing T1-images is small, T2-errors are large, for all tissues studied. This is also e.g. visible in Fig 6, second subject, right image, where T2w-contrast is lacking in the center-top-part. This raises the question if T2w-contrast can be inferred from short TE T1w-images. Please discuss as a limitation of this work."
>
> A figure is added to the paper that showcases the example mentioned by the reviewer. Although the T2 maps are generally smaller than expected, the predicted signal is impressive and by adding the baseline errors to Figure 2., the improvements made by the model over the baseline error are easier to interpret. A paragraph in the discussions section is now dedicated to the differences between the quantitative maps of the LSQ and the model.
>
>
> "Minimal prior work on the field is mentioned, domain transfer learning approaches have been proposed before. The novelty of incorporating the signal acquisition equation and using unsupervised learning should be evaluated against relevant work."
>
> The recommended articles are very much appreciated, and they were used to revise the introduction into the topic. However evaluating against their methods are out of the scope of this paper.
>
>
> "The authors can be more precise in their terminology of multi-contrast data through the text. It is mentioned that the train and val sets were based on multi-contrasts scans (from all five contrasts), followed by the remark that given the self-supervising aspect of the method, no multi-contrast scans were required for training."
>
> To emphasize the significance of the fact that no paired data from different contrasts is required for training, the authors revised the description of the training dataset. Contrast pairs are only required for the evaluation of the model. The confusion may derive from the fact that "To speed up the data acquisition process, every patient was scanned for all 5 contrasts independently" but to make sure no paired information is present in the training data, "all slices were shifted randomly in both directions by up to 10 pixels".
>
>
> "Fig. 2 represents a contrast in orange color, whereas in the text it is mentioned as black."
>
> The main text was updated to refer to the correct color, "orange".
>
>
> "The publicly available materials provide a viewer for three given images instead of, e.g., gifs, with the option to change contrast settings. There is no real value of the publicly available materials since no code is available, neither can be extended on other data using the pretrained model."
>
> For an easy evaluation of the model, the authors published a tool that implements the pre-trained model (contrast_transfer_keras.h5). The tool does not require any programming skills, and can easily test the model for the three example images. The tool is not limited to the three images, however it is limited to .jpeg images of size 256x256. If the user wishes to investigate the model further and evaluate it on other datasets, they are encouraged to do so using the pre-trained model itself, named ``contrast_transfer_keras.h5''. To avoid confusion about model availability and transparency (which the authors find very important), the corresponding description was updated in the final paragraph of the Conclusions of the paper.
>
>
> "Fig. 6 is hard to interpret, please reduce margins. The second row should be replaced by plotting the difference between the target and the predicted image. The bottom row should contain pairs of the predicted quantitative maps and the LSQ ground truth maps. Also adjust the range, removing peak intensities, and consider saturating the bladder (T1/T2 are difficult to estimate in free water)"
>
> The margins of Figure 6 have been reduced, a fourth example is introduced  and the images are now distributed over two pages to increase figure sizes. One example is also introduced in the main text for visual assessment. Although seeing the errors between the predictions and the ground truth is certainly useful, the second row was kept to show the predictions of the model, as the authors believe this visualization of the predictions is essential for the reader to get a grasp of how the model performs. For quantitative evaluations, to get a grasp of the errors, the reader can turn to Table 2.

---

### Official Review · AnonReviewer1 · 2021-03-09

**Confidence:** 5
**Preliminary Rating:** 3
**Recommendation:** Poster

**Summary:**

This paper presents a deep generative network that is trained to transfer any MRI contrast to any other. This method relies on the equation relating signal contrast to three quantitative properties of physical tissues, namely proton density PD, T1 and T2 relaxation times as well as two acquisition parameters, the echo and repetition times, TE et TR. The model takes as input one MR slice and the targeted values of TE and TR and outputs corresponding MR slice as the input image but with the targeted contrast. The network consists of a UNET-based architecture for the generator part and a patch-GAN discriminator. This model is evaluated on 100 patient exams, each constituted of 5 MRI pelvic acquisitions, acquired for different couples of TE and TR values. This model is shown to produce accurate results

**Strengths:**

-Well written
-Computation of baseline performance, if I understand it correctly, is interesting. It indeed shows that the different metrics (SSIM, PSNR) produce very good metrics (eg SSIM=0.961) when computed on pairs of MR images with different contrast. This allows moderating conclusions that can be drawn from the performance of the synthesis network (SSIM  0.974).
-The architecture of the generative models allows generating the three physical parametric maps, T1, T2 and PD, which might be of interest
-Code is available on zenodo


**Weaknesses:**

-Lack of visual assessment of the generated synthetic data

-SOTA on deep synthesis architectures should be updated with other efficient architecture including cycle-GAN, for instance, and further detailed to justify the choice of the deep generative architecture considered in this study.

-Figure 6  is part of the Appendix and refered to as 'additional images' while it is the only and essential illustration of the synthetic image produced by the model. This figure should be part of the main paper. In this form, i am not sure the paper format respects MIDL rules.


**Deanonymize Review:**

no

**Detailed Comments:**

-The choice of the deep generative architecture considered in this study should be motivated. Did the author consider alternate models, such as Cycle-GAN models?

-Some discussion on the hyperparameter settings (eg patch size for CycleGAN) should be added, especially to evaluate their impact on the performance metric or robustness of the model.

-Some discussion should be added on the comparison of the baseline metrics (eg SSIM=0.961) with those achieved with the synthetic networks (eg SSIM  0.974). The authors could indeed apply some statistical tests to estimate if such a difference is statistically significant.

-In Appendix B, Figure 5 should be clarified. Please a scale and title the y-axis. The red area enlarges with increasing number of epochs, meaning that the discriminator keeps improving. Analysis of the blue area is less clear, it seems to slightly decrease over time, how should this be interpreted? Please clarify.

-Appendix C : images are too small. These are not ‘additional’ but the only provided images. Please add some illustrations in the main paper not in the appendix.



**Justification Of The Preliminary Rating:**

This paper presents an interesting application of deep generative models for the synthesis of multi-contrast MRI images conditioned to TE and TR acquisition parameters. The paper would gain soundness by motivating the choice of the network architecture and optimizing the hyperparameters as well as by emphasizing the visual quality of the generated synthetic data.

**Paper Type:**

both

**Questions To Address In The Rebuttal:**

Please address all detailed comments above.

**Special Issue:**

no

---

> ### Author Response · Authors · 2021-03-16
> **Response to review #1**
>
> "Lack of visual assessment of the generated synthetic data"
>
> We address this by including an example in the main text showing an input image contrast, a specific target contrast, and the corresponding predicted image. The figure also shows the predicted PD, T1 and T2 maps. The results are evaluated in the Discussion. Additional examples on Figure 6 remain in the appendix, but all assessment is done on information contained in the main text.
>
>
>  "SOTA on deep synthesis architectures should be updated with other efficient architecture including cycle-GAN, for instance. [...]"
>
> The authors agree with the reviewer on the importance of CycleGANs in current research, therefore the introduction to the available deep learning methods is updated accordingly. However the current study put emphasis on easy reproducibility on new data with a simple way to adjust the number of contrasts included in the training process. As for CycleGANs, we state in the introduction that "increasing the number of supported contrasts to transfer to also increases the complexity of the problem by the number of models to train", therefore we decided not to use a CycleGAN setup but to introduce our own simple conditional GAN. Hence, for retraining on new data, the user only has to adjust the number of output classes of the discriminator to the number of contrasts included in the training. Based on the presented results, the authors have decided to continue the study on a larger and more diverse dataset, and possibly explore other GAN architectures therefore the reviewer's comments are very appreciated.
>
> "Figure 6 is part of the Appendix and refered to as 'additional images' while it is the only and essential illustration of the synthetic image produced by the model. [...] In this form, I am not sure the paper format respects MIDL rules."
>
> A figure visualizing the results is now included in the main text. The visual assessment is done on this example, however four more examples remain in the appendix for additional visualization of the results. Since these figures are not included in drawing the conclusions of the method, the authors hope that this new outline respects the rules set by the conference.
>
>
> " [...] Did the author consider alternate models, such as Cycle-GAN models?"
>
> A paragraph in the Introduction now describes CycleGANs and why they were not chosen for this study.
>
>
> "Some discussion on the hyperparameter settings (eg patch size for CycleGAN) should be added [...]"
>
> The patch size of the PatchGAN, kernel sizes of the convolutional layers, the standard deviation of the Gaussian noise layers used, the optimizers for training and learning rates for both the generator and discriminator, the losses used for both models, normalizing the weights, and the number of updates of the discriminator before updating the adversarial network were all tuned before selecting the final model, however due to page and time constraints, evaluating the robustness of the model against each of these parameters was outside the scope of this paper.
>
>
> "Some discussion should be added on the comparison of the baseline metrics (eg SSIM=0.961) with those achieved with the synthetic networks (eg SSIM 0.974).[...]"
>
> The distribution of the NRMSE errors was updated to contain the baseline error for each input-target contrast as well, which helps with determining the significance of the improvements for each case. The authors believe the changes highly improved the evaluation of the method. The discussion for this evaluation was updated accordingly.
>
>
> "In Appendix B, Figure 5 should be clarified. [...]"
>
> The figure describes the competition between the discriminator and the adversarial network. "After each epoch the performance of the discriminator network is evaluated through the accuracy of the discriminator for classifying generated images as fake (red line, D fake accuracy), while the performance of the adversarial network was evaluated by the accuracy of the discriminator for classifying generated images as the contrast they belong to (blue line, GAN accuracy). The white lines show where images were classified as real but from the incorrect contrast''. The y-axis is now titled, and its scale is discussed in the description of the figure, stating that ``For each epoch the length of these three lines add up to 100\%''. Although the accuracy of the discriminator noticeably increases over time, it never dominates over the adversarial network. Both the main text of Appendix B, and the figure caption were revised for clearer interpretation.
>
>
> "Appendix C : images are too small. These are not ‘additional’ but the only provided images. [...]"
>
> An example is now included in the main text for visual assessment. The additional image sizes were increased, an extra example is added and the figure is distributed over 2 pages.

---

### Author Response · Authors · 2021-03-16
**Official comment by authors**

The authors believe that addressing the weaknesses of the paper by the reviewers has greatly benefited the quality of the paper. For this they would like to thank the supporting and thorough comments of the reviewers.

There were three major updates to the paper:
 - A visual assessment was added to the main text, so the reader does not have to refer to the appendix to find information essential to the conclusions of the paper.
 - The boxplots showing the distribution of the prediction error between input and target contrasts are now extended to include the baseline errors as well. This addition has introduced an extremely useful way to evaluate the method.
 - The original evaluation of the decision boundaries of the discriminator included more information than what was meaningful. A reviewer has pointed this out, and therefore we have revised the evaluation of the model to only focus on the parameter space that adds great value to the evaluation of the method.


These updates increased the length of the paper therefore to fit the page constraints, several sentences had to be re-written to be more concise. Hence, the authors highlight only the major changes in the revised paper.

---

### Meta-Review · Area_Chair1 · 2021-03-24

**Recommendation:** Accept (Poster)

**Metareview:**

This paper presents a deep generative models for  multi-contrast synthesis MRI. Th proposed unsupervised method uses knowledge of the MR physics models and is applied to radiotherapy. Most of the reviewers found this paper interesting. The technical choice and the validation could be improved, especially the choice of the baseline method.

**Paper Type:**

methodological development

---

### Decision · Program_Chairs · 2021-03-31

Accept